# Attenuation of Myocardial Dysfunction in Hypertensive Cardiomyopathy Using Non-R-Wave-Synchronized Cardiac Shock Wave Therapy

**DOI:** 10.3390/ijms232113274

**Published:** 2022-10-31

**Authors:** Fei Li, Zhe Zhen, Si-Jia Sun, Yu Jiang, Wei-Hao Liang, Markus Belau, Rafael Storz, Song-Yan Liao, Hung-Fat Tse

**Affiliations:** 1Cardiology Division, Department of Medicine, Queen Mary Hospital, The University of Hong Kong, Hong Kong SAR, China; 2Department of Cardiology, The First Affiliated Hospital of Sun Yat-sen University, Guangzhou 510060, China; 3Storz Medical AG, 8274 Tägerwilen, Switzerland; 4Shenzhen Institutes of Research and Innovation, The University of Hong Kong, Hong Kong SAR, China; 5Hong Kong-Guangdong Joint Laboratory on Stem Cell and Regenerative Medicine, The University of Hong Kong and Guangzhou Institutes of Biomedicine and Health, Guangzhou 510530, China; 6Center for Translational Stem Cell Biology, Hong Kong SAR, China; 7Cardiac and Vascular Center, The University of Hong Kong-Shenzhen Hospital, Shenzhen 518053, China

**Keywords:** cardiac shock wave, hypertension, cardiomyopathy

## Abstract

Cardiac shock wave therapy (CSWT) is a novel therapeutic procedure for patients with angina that is refractory to conventional therapy. We investigated the potential mechanism and therapeutic efficacy of non-R-wave-triggered CSWT to attenuate myocardial dysfunction in a large animal model of hypertensive cardiomyopathy. Sustained elevated blood pressure (BP) was induced in adult pigs using a combination of angiotensin-II and deoxycorticosterone acetate (DOCA). Two sessions of non-R-wave-triggered CSWT were performed at 11 and 16 weeks. At 10 weeks, systolic and diastolic blood pressure, LV posterior wall thickness and intraventricular septum thickness significantly increased in both the hypertension and CSWT groups. At 20 weeks, +dP/dt and end-systolic pressure-volume relationship (ESPVR) decreased significantly in the hypertension group but not the CSWT group, as compared with week 10. A significant improvement in end-diastolic pressure-volume relationship (EDPVR) was observed in the CSWT group. The CSWT group exhibited significantly increased microvascular density and vascular endothelial growth factor (VEGF) expression in the myocardium. Cytokine array demonstrated that the CSWT group had significantly reduced inflammation compared with the hypertension group. Our results demonstrate that non-R-wave-triggered CSWT is safe and can attenuate LV systolic and diastolic dysfunction via enhancement of myocardial neovascularization and anti-inflammatory effect in a large animal model of hypertensive cardiomyopathy.

## 1. Introduction

Heart failure (HF) represents a major socioeconomic health burden around the world, and approximately half of affected patients suffer from HF with preserved ejection fraction (HFpEF). In China, up to 1.3% of the population suffers from HF. The prevalence of HF in the United States and Europe varies from 1% to 12% based on different studies or reports [1]. Hypertensive cardiomyopathy with left ventricular hypertrophy (LVH) is the most common etiology of HFpEF. Although early and effective treatment of hypertension can prevent the development of LVH, current pharmacological therapies such as β-blocker and renin-angiotensin-aldosterone system blockers fail to address the underlying pathophysiological hallmarks of HFpEF. The latter include impaired left ventricular relaxation and increased diastolic stiffness associated with maladaptive LVH and myocardial fibrosis [2,3]. Preclinical studies suggest that coronary microvascular dysfunction (CMD) associated with hypertension plays an important role in the development of HFpEF [4]. Indeed, recent clinical studies demonstrate that up to 75% of HFpEF patients suffer from CMD despite the absence of significant coronary artery disease [5]. Therefore, CMD is a potent novel target for treatment of HFpEF. However, the current therapeutic approach to CMD is limited.

Ultrasound-guided cardiac shock wave therapy (CSWT) has been proposed as a therapeutic option for patients with angina that is refractory to conventional medical treatment [6,7]. In contrast to high-energy extracorporeal shock wave therapy for the treatment of urinary tract stones [8], CSWT is based on the non-invasive application of low-intensity shock waves to the ischemic myocardium. This can increase nitric oxide synthesis, vascular endothelial growth factor (VEGF) expression, differentiation of bone marrow cells into endothelial cells, and the amount of circulating endothelial progenitor cells to enhance angiogenesis and reduce the inflammatory response, oxidative stress, cellular apoptosis and fibrosis [9,10,11,12,13]. Currently, CSWT is delivered using a special applicator through the acoustic window to the treatment area under electrocardiographic R-wave gating to avoid potential proarrhythmias, but it’s clinical feasibility is limited by the need for prolonged treatment [14]. The use of non-R-wave-triggered CSWT should significantly reduce the duration of treatment but its safety remains unclear.

We hypothesize that non-R-wave-triggered CSWT can be delivered without proarrhythmias and can improve CMD via neovascularization and anti-inflammatory effect, with consequent attenuation of myocardial dysfunction in hypertensive cardiomyopathy.

## 2. Results

### 2.1. CSWT Has No Effect on Blood Pressure in Hypertensive Cardiomyopathy

No significant differences in ambulatory systolic or diastolic BP were observed from baseline to 10- and 20-week follow-up in the control group (Figure 1A,B). After induction of hypertension, a significant increase of systolic and diastolic BP level was observed in the hypertension group and CSWT group at 10 and 20 weeks (Figure 1A,B). This result demonstrated that severe hypertension had been established in porcine of both the hypertension and CSWT groups. At 20 weeks, there were no significant differences in systolic or diastolic blood pressure between the hypertension and CSWT groups (Figure 1A,B).

### 2.2. Hemodynamic Data

LV pressure volume loop analysis was used to perform hemodynamic assessments to evaluate cardiac function at baseline, 10 weeks and 20 weeks in the control, hypertension and CSWT groups (Figure 2). Maximum rate of LV pressure rise (+dP/dt) and end-systolic pressure volume relationship (ESPVR) were calculated to assess LV systolic function. Maximum rate of LV pressure reduction (−dP/dt) and end-diastolic pressure volume relationship (EDPVR) were calculated to assess LV diastolic function. In the control group, there was no significant difference of +dP/dt, ESPVR, −dP/dt and EDPVR from baseline to 10- and 20-week follow-up (Figure 2A–D). At 10 weeks, although no significant differences of +dP/dt and −dP/dt were observed in the hypertension and CSWT groups (Figure 2A,B), the ESPVR and EDPVR significantly increased in the hypertension and CSWT groups compared with baseline (Figure 2C,D). Compared with 10 weeks, there was a significant decrease of +dP/dt and ESPVR in the hypertension group at 20 weeks, but not in the CSWT group (Figure 2A,C). Moreover, CSWT significantly decreased EDPVR in the CSWT group at 20 weeks compared with 10 weeks (Figure 2D).

### 2.3. Echocardiographic Data

Echocardiographic measurement was performed at baseline, 10 weeks and 20 weeks to evaluate cardiac function and ventricular wall thickness (Figure 3). There was no significant difference of LVEF in the control and CSWT groups from baseline to 10- and 20-week follow-up (Figure 3A). A significant decrease of LVEF was observed at 20 weeks compared with 10 weeks in the hypertension group (Figure 3A). LV posterior wall thickness, interventricular septum thickness and LV mass index significantly increased in the hypertension and CSWT groups following induction of hypertension at 10 and 20weeks compared with that at baseline and in the control group, respectively (Figure 3B–D). Neither LV end systolic dimension nor LV end diastolic dimension have significant difference during the 20-week follow-up among the control, hypertension and CSWT groups (Figure 3E,F).

Taken together, our study showed that CSWT neither decreased systolic nor diastolic BP nor did it reduce LVH, but it significantly attenuated cardiac dysfunction in hypertensive cardiomyopathy.

### 2.4. No Incidence of Arrhythmias during CSWT

In the CSWT group, all animals underwent non-R-wave-triggered CSWT without any complications. Continuous ECG monitoring during CSWT revealed only 8 single premature ventricular contractions out of a total 80,000 applications of CSWT, and no sustained arrhythmias were noted during CSWT. Moreover, there was no significant change in serum cardiac troponin level after CSWT (Appendix A, *p* > 0.05).

### 2.5. Histological and Immunohistochemical Data

Fibrosis level was detected by Masson’s trichrome staining for the three groups at the 20th week (Figure 4A). A significant increase in area-percentage of fibrosis at the endocardium and mid-myocardium was observed in the hypertension group and CSWT group compared with the control group (Figure 4B,C). No significant difference in the degree of fibrosis was observed in the epicardium among the three groups (Figure 4D).

To determine whether neovascularization contributed to the functional improvements observed in the CSWT group, immunohistological staining with alpha-smooth muscle actin (α-SMA) antibodies was performed to determine vessel formation in the endocardium, mid-myocardium and epicardium (Figure 5A). Compared with the control group, there were no significant differences in positive staining for SMA-positive microvascular density in the hypertension group among three layers of myocardium (Figure 5B–D). On the contrary, SMA-positive microvascular density significantly increased in the CSWT group compared with the control and hypertension groups in all three layers of myocardium (Figure 5B–D). Furthermore, Western blot analysis revealed local VEGF expression in the myocardium was increased significantly in the CSWT group compared with the control and hypertension groups (Figure 5E,F).

### 2.6. Cytokines Array Data

The level of circulating cytokines was detected using cytokine array in the control, hypertension and CSWT groups at the 20th week (Figure 6). After induction of hypertension, expression of IL-1β, IL-6, IL-4, TGFβ1, TNFα and GM-CSF significantly increased in the hypertension group compared with the control group, and the level of expression of IL-10, IL-8, IL-12 and IFNg was not detected in the control group (Figure 6A–J). Expression of IL-10, IL-8, IL-1β, IL-6, IL-4, TGFβ1, TNFα and GM-CSF significantly decreased in the CSWT group compared with the hypertension group (Figure 6A–H). There was no significant difference of expression of IL-12 and IFNg between the hypertension group and CSWT group. (Figure 6I,J).

## 3. Discussion

To the best of our knowledge, this is the first pre-clinical study to investigate the safety and efficacy of CSWT in a clinically relevant large animal model of hypertensive cardiomyopathy. First, we demonstrated that non-R-wave-triggered CSWT can be safely delivered without any proarrhythmias or myocardial injury. Second, although there were no significant therapeutic effects on BP, LVH or myocardial fibrosis, our results showed that CSWT could significantly preserve LV systolic function and attenuate LV diastolic dysfunction in hypertensive cardiomyopathy. Third, CSWT significantly increased microvascular density via local VEGF expression and exerted systemic anti-inflammatory effects. These findings provide important proof-of-principle data to support the application of non-R-wave-triggered CSWT as a novel non-invasive therapy for hypertensive cardiomyopathy.

Hypertensive cardiomyopathy with left ventricular hypertrophy is the most common etiology of HFpEF [15]. Unfortunately, the therapeutic approach to improve the clinical outcomes in HFpEF is limited once the disease is established. Currently, only treatment with sodium–glucose cotransporter 2 inhibitors has been shown to reduce the risk of cardiovascular death or hospitalization for heart failure in HFpEF [16]. Nonetheless, HFpEF is a heterogeneous disease with diverse mechanisms and abnormalities [3,4,17,18]. Other than the well-known hemodynamic and structural changes with LV diastolic dysfunction, and LV hypertrophy and fibrosis, respectively, CMD and systemic microvascular inflammation have also been recognized as important mechanisms for HFpEF [17,18]. In the PROMIS-HFpEF study, up to 75% of patients with a clinical diagnosis of HFpEF had CMD [5]. Hence, CMD is a potential therapeutic target in HFpEF. In this study, we aimed to determine whether CSWT could serve as a novel non-invasive therapy in a large animal model of hypertensive cardiomyopathy with phenotypes of HFpEF [19].

Shockwaves have been applied to clinical patients during lithotripsy as a way to disintegrate kidney and urethral stones and enhance healing of soft-tissue defects of non-healing wounds, but also as an effective and non-invasive therapeutic strategy for ischemia heart disease, such as refractory angina and ischemic cardiomyopathy [6,7,8,20,21]. CSWT delivers low-intensity shockwaves directly onto the myocardium, inducing the release of angiogenic factors, especially VEGF, to enhance neovascularization and reduce inflammatory response [9,10,11,12,13,20]. Recent clinical studies have demonstrated that CSWT increased biomarkers of angiogenesis and decreased those of inflammation in patients with coronary artery disease and HF [22]. As expected, two sessions of CSWT had no significant effect on BP and thus had none on LV structural changes, including LVH and myocardial fibrosis, which are associated with persistent elevated BP. Nonetheless, CSWT induced global LV myocardial neovascularization, as evidenced by increased capillary density in all three layers of the myocardium via upregulation of VEGF expression, as shown in prior studies [13,23,24]. Moreover, CSWT also had potent systemic anti-inflammatory effects in hypertensive cardiomyopathy, as demonstrated by the significant reduction in several proinflammatory cytokines, including IL-10, IL-8, IL-1β, IL-6, IL-4, TGFβ1 and TNFα, in this study. Prior studies showed that CSWT improved cardiac angiogenesis in ischemic cardiac disease without any increase of serum Troponin I level and cardiac arrhythmic risk [23]. Meanwhile, in normotensive animals, there is no evidence that shockwave therapy induces cellular damage and inflammatory response in the myocardium or changes the ultrastructure morphology, blood pressure and cardiac ventricular function when used within a therapeutic range [25]. Our results support these findings and further demonstrate that CSWT does not cause any complication in hypertensive cardiomyopathy. Our study showed that no increase of blood pressure, no increase of serum Troponin T, no cardiac arrhythmia or cardiac sudden death were observed. More importantly, the myocardial angiogenic and anti-inflammatory actions of CSWT were associated with preservation of LV systolic function and improved LV diastolic function in our large animal model of hypertensive cardiomyopathy.

### 3.1. Perspectives

Our findings have several potential clinical implications. First, our results provide novel evidence that treatment of CMD via myocardial neovascularization using CSWT, independent of the persistence of hemodynamic stress with elevated BP and LVH, can still improve LV function. Second, the findings in this preclinical large animal study provide important proof-of-principle data to support future clinical trials of CSWT in patients with HFpEF. Third, we also demonstrated that non-R-wave-triggered CSWT was safe, with no proarrhythmia or myocardial injury in the presence of HF. Previous preclinical studies demonstrated that CSWT was a new therapy method for patients who suffered from chronic myocardial ischemia or refractory angina unresponsive to conventional treatments such as percutaneous coronary intervention (PCI) or coronary artery bypass grafting (CABG) [6]. According to the above results, our findings support previous studies that showed application of CSWT to induce angiogenesis and anti-inflammation to ameliorate CMD, which is a potential therapeutic target in HFpEF. Therefore, patients who suffer from hypertensive cardiomyopathy with HFpEF could benefit from CSWT. This should further enhance the clinical application of CSWT for different cardiac diseases by significantly shortening treatment duration.

### 3.2. Limitations

Our study has several limitations. First, we did not include a treatment arm with anti-hypertensive therapy and CSWT in this study. It remains unclear whether CSWT can provide an incremental benefit on top of BP lowering in our animal model of hypertensive cardiomyopathy. Our recent study showed that BP lowering with splanchnic denervation could also result in regression of LVH and improved LV diastolic function in the same animal model of hypertensive cardiomyopathy [19]. Second, we did not measure coronary microvascular function in this study, as measurement of direct invasive coronary flow reserve was not possible in our animal lab, and measurement of indirect coronary flow reserve using the adenosine stress transthoracic Doppler was not feasible due to the poor echo window in swine. Third, the mechanisms of systemic anti-inflammatory effects of CSWT observed in this system remain unclear. Finally, it remains unclear whether our findings regarding the treatment effects of CSWT in hypertensive cardiomyopathy would translate to a more heterogeneous phenotype of HFpEF.

In conclusion, non-invasive, non-R-wave-triggered CSWT significantly improved the microvascular density in the myocardium, ameliorated systemic inflammation and improved cardiac performance without any effect on BP and LV hypertrophy or fibrosis in a large animal model of hypertensive cardiomyopathy. The safety and efficacy of CSWT in hypertensive cardiomyopathy with HFpEF should be further investigated in clinical trials.

## 4. Methods and Materials

### 4.1. Animal Model of Hypertensive Cardiomyopathy

All procedures that involved animals were approved by the Committee on the Use of Live Animals in Teaching and Research (CULATR) at the University of Hong Kong. (CULATR number 4662-18). Adult female pigs weighing 35–45 kg (9–12 months old) were used in this study to create a large animal model of hypertensive cardiomyopathy as described in our previous studies [19]. In brief, all animals underwent implantation of an osmotic infusion pump (Tricumed Medizintechnik GmbH, Kiel, Germany), Physiotel telemetry (Data Sciences International) and implantation of subcutaneous deoxycorticosterone acetate (DOCA) pellets using sterile surgical techniques. The pump was primed with 10 mg of angiotensin-II (Ang-II, Sigma-Aldrich, St. Louis, MO, USA) in 0.4 mL of sterile isotonic saline, infused at a constant rate of 0.6 µL/h (i.e., 0.015 mg/h) and replaced every 4 weeks. For implantation of the telemetry unit, a fluid-filled catheter was inserted into the carotid artery and connected to a telemetry device (Data Sciences International, Saint Paul, MN, USA) located in the body of the implant. The implant transmitted blood pressure (BP) data telemetrically to a receiver that converted the radiofrequency signal to a PC-based data collection system. In addition, a subcutaneous pocket was created over the dorsal neck region for implantation of the DOCA pellets (Innovative Research of America, Sarasota, FL, USA) to release 100 mg/kg DOCA over a period of 12 weeks.

Telemetry recording of ambulatory BP (systolic, diastolic and mean arterial pressure) was performed every 2 weeks in conscious animals. Serial echocardiogram and invasive hemodynamic assessment were performed at baseline, 10 weeks and 20 weeks in all three groups to assess LV dimension and function, respectively. All animals were sacrificed for histological and immunohistochemical assessments at 20 weeks.

### 4.2. Cardiac Shock Wave Therapy

At 10 weeks following Ang-II infusion and DOCA implantation, animals with hypertensive cardiomyopathy were assigned to the no-therapy group (hypertension group, n = 8) or CSWT group (n = 8). Six pigs without hypertension induction served as the control group (n = 6). Animals in the CSWT group received two sessions of CSWT, in the 11th and 16th weeks, using a non-R-wave-triggered fixed frequency at a constant frequency of 4 Hz, with the guidance of echocardiography instruments mounted to the basal septal, anterior, inferior and lateral wall. In each session, 400 shots/spot were applied to 10 spots at energy level 3 (Modulith SLC, STORZ MEDICAL AG, Tägerwilen, Switzerland). During the procedure, the safety of CSWT, including the incidence of premature atrial and ventricular contractions and their temporal correlation to the shockwave pulse were analyzed, and pigs were monitored for the occurrence of severe arrhythmia such as ventricular fibrillation.

### 4.3. Echocardiographic Examination

All animals underwent transthoracic echocardiograms using a commercially available echocardiographic system with a 3–9 MHz transducer (Vivid q, GE Vingmed, Horten, Norway) at baseline and in the 10th and 20th weeks [19]. Standard two-dimensional and M-mode measurements were performed to evaluate LV ejection fraction (LVEF), LV posterior wall thickness, interventricular septum thickness, LV mass index and LV end-systolic and end-diastolic dimension. All echocardiographic measurements were interpreted in a blind fashion and at least three consecutive beats were analyzed offline using a computer workstation (Echo Pac, GE Vingmed, Horten, Norway, version 110).

### 4.4. Invasive Hemodynamic Assessment

Invasive hemodynamic assessment was performed to evaluate cardiac function at baseline and in the 10th and 20th weeks [23,24]. In brief, a 7-Fr combined catheter-micromanometer (Millar instruments, Houston, TX, USA) was advanced into the LV apex via the femoral or carotid artery. A cardiac function analyzer (CFL 512, CD Leycom) was then connected to the catheter. LV positive and negative dP/dt_max_, end-systolic pressure–volume relationship (ESPVR) and end-diastolic pressure–volume relationship (EDPVR) were recorded to assess LV performance.

### 4.5. Histological Assessment

Detailed histological assessments were performed to evaluate changes in cardiac structure and the myocardial vascular density [26,27]. In brief, hearts were harvested after all animals were sacrificed at the 20th week. Hearts were then fixed with 4% paraformaldehyde, embedded in paraffin and cut into 4-μm slides. The degree of fibrosis was detected by Masson’s trichrome staining using a commercially available kit (AACSC009, American Master Tech Scientific, CA, USA). Immunofluorescence staining with anti-porcine α-smooth muscle actin (SMA, 1:100, ab5694, Abcam, MA, USA) was performed following a standard protocol to evaluate vascular density. The images were captured by AxioVision Rel. 4.5 software (Zeiss, GmbH, Oberkochen, Germany) and the area-percentage of positive staining was calculated using image J software (National Institutes of Health, Bethesda, MD, USA). All data were measured in five random 20× fields in a blinded fashion.

### 4.6. Western Blot Analysis

Total protein samples were extracted from hearts and loaded on SDS-PAGE gel and then transferred to PVDF membranes. After blocking with BSA for 2 h, the membranes were incubated overnight with VEGF monoclonal antibody (1:1000, VG76e, Thermo Fisher Scientific, Waltham, MA, USA) at 4 °C. Membranes were then incubated with appropriate fluorescence conjugate secondary antibody for 2 h at room temperature. Finally, the membranes were visualized by enhanced chemiluminescence and protein bands were quantified using Image J 1.8.0 (Image J Software, NIH, Bethesda, MD, USA).

### 4.7. Cytokine Array

At 20 weeks, blood samples were collected from the femoral vein and centrifuged at 6000× *g* for 15 min to collect plasma. Level of plasma porcine cytokines, including interleukin (IL)-10, IL-8, IL-1β, IL-6, IL-4, IL-12, transforming growth factor β1 (TGFβ1), tumor necrosis factor (TNF-α), interferon gamma (INFg) and granulocyte–macrophage colony-stimulation factor (GM-CSF) were determined using a cytokine array kit (GSP-CYT-1, RayBiotech, Peachtree Corners, GA, USA) following a standard protocol. In brief, 100 μL of plasma was added to each well and incubated at room temperature for 2 h. After washing, the slide was incubated with biotinylated antibody cocktail for 2 h, and then further incubated with Cy3 equivalent dye-labeled streptavidin for 1 h. Finally, a laser scanner equipped with a Cy3 wavelength was used to scan the signals.

### 4.8. Biomarker Assessment

In the CSWT group, plasma porcine cardiac Troponin T (MyBioSource, San Diego, CA, USA) level was evaluated before and 1 day after the first session of CSWT using an enzyme-linked immunosorbent assay (ELISA) kit (Troponin T, ab223360, Abcam, Waltham, MA, USA) to determine myocardial injury.

### 4.9. Statistical Analysis

All data are expressed as mean ± SEM. Comparison of serial changes in blood pressure, echocardiographic and invasive hemodynamic parameters at different time points between groups was made using a two-way repeated ANOVA with Tukey’s test as appropriate. Histology and immunohistochemical staining results of different groups were compared by independent Student *t*-tests. Statistical significance was defined as a *p*-value less than 0.05. All statistical analyses were performed using SPSS 23, (SPSS software, Inc., Chicago, IL, USA).

## Figures and Tables

**Figure 1 ijms-23-13274-f001:**
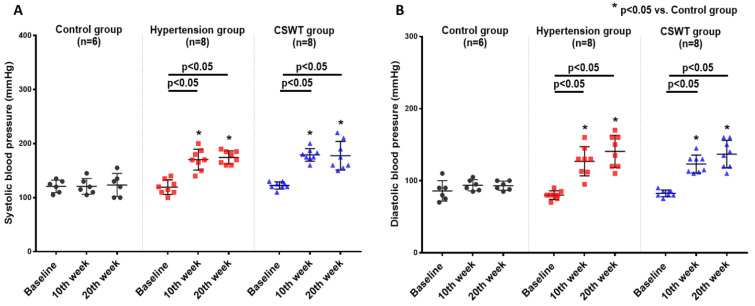
CSWT (cardiac shock wave therapy) effect on blood pressure in hypertensive cardiomyopathy. Serial changes to ambulatory blood pressure at baseline, 10th week and 20th week follow-up in control, hypertension and CSWT groups. (**A**). 10 weeks after induction of hypertension, systolic blood pressure of animals significantly increased in hypertension and CSWT groups compared with control group. At the 20th week, CSWT did not reduce the systolic blood pressure compared with hypertension group. (**B**). Diastolic blood pressure in hypertension and CSWT groups was particularly elevated at the 10th week after induction of hypertension. At the end of the experiment, no significant decrease of diastolic blood pressure was observed in CSWT group compared with hypertension group. Two-way repeated ANOVA with Tukey’s test was used, * *p* < 0.05 versus control group.

**Figure 2 ijms-23-13274-f002:**
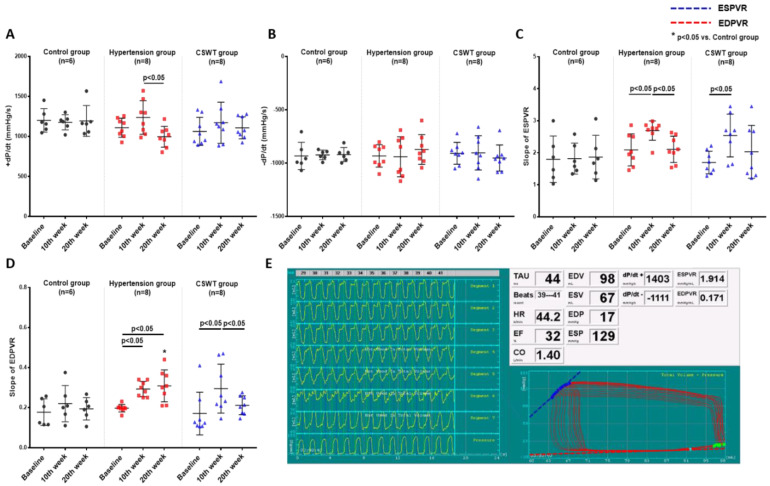
CSWT effect on hemodynamic parameters. Hemodynamic parameters at baseline, 10th week and 20th week follow−up in control, hypertension and CSWT groups. (**A**). No significant difference of +dP/dt was observed in hypertension and CSWT groups at the 10th week compared with control group and at baseline. At the 20th week, +dP/dt decreased significantly in hypertension group compared with the 10th week. (**B**). No significant difference of −dP/dt among the three groups throughout whole experiment. (**C**). At the 10th week in hypertension and CSWT groups, ESPVR (end−systolic pressure−volume relationship) increased compared with baseline. At the 20th week, there was significant decrease of ESPVR in hypertension group but not in CSWT group compared with the 10th week. (**D**). 10 weeks after induction of hypertension, EDPVR (end−diastolic pressure−volume relationship) increased significantly in hypertension and CSWT groups compared with baseline. At the 20th week, CSWT significantly reduced EDPVR in CSWT group compared with the 10th week. (**E**). Representative tracing of the left ventricular pressure volume loop. Two−way repeated ANOVA with Tukey’s test was used, * *p* < 0.05 versus control group.

**Figure 3 ijms-23-13274-f003:**
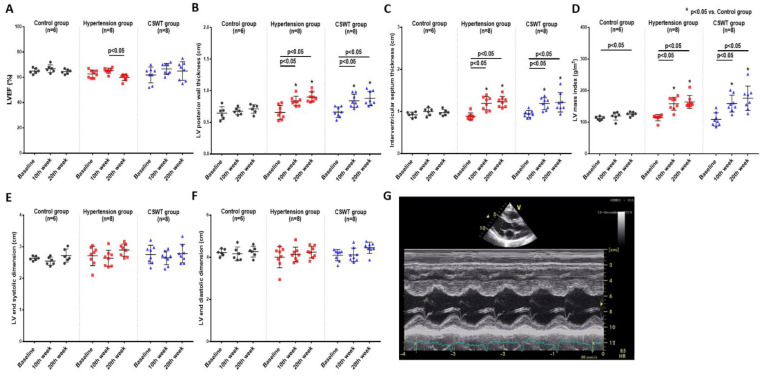
Echocardiographic data. Echocardiographic parameters at baseline, 10th week and 20th week follow−up in control, hypertension and CSWT groups. (**A**). No significant difference of LVEF (left ventricular ejection fraction) was observed in control and CSWT groups throughout the whole experiment. At the 20th week, LVEF decreased in hypertension group compared with that at the 10th week. (**B**,**C**). After induction of hypertension, LV (left ventricle) posterior wall thickness and interventricular septum thickness increased significantly in hypertension and CSWT groups at the 10th week and 20th week compared with those at baseline and in the control group. (**D**). LV mass index increased significantly in hypertension and CSWT groups compared with that of control group at the 10th week and 20th week. (**E**,**F**). No significant difference of LV end systolic dimension and LV end diastolic dimension were observed among the three groups throughout the whole experiment. (**G**). Representative echocardiographic M−mode tracing. Two−way repeated ANOVA with Tukey’s test was used, * *p* < 0.05 versus control group.

**Figure 4 ijms-23-13274-f004:**
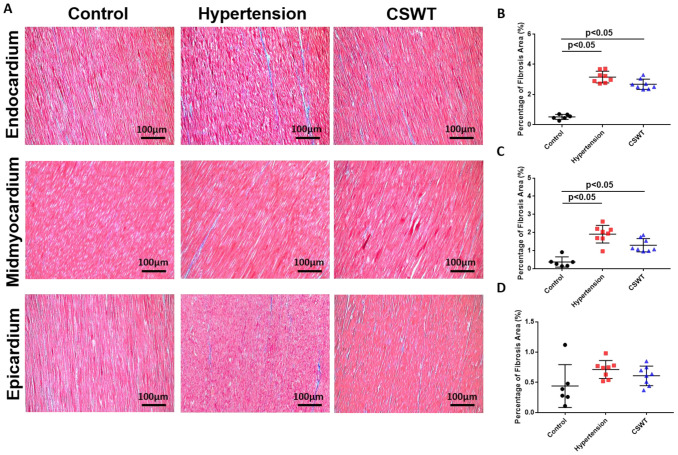
CSWT effect on fibrosis level in hypertensive cardiomyopathy. Masson’s trichrome staining of heart sections in the three groups at the 20th week. (**A**). The degree of fibrosis in endocardium, midmyocardium and epicardium were assessed by Masson’s trichrome staining. (**B**). In endocardium, percentage of fibrosis area in hypertension and CSWT groups increased significantly compared with control group. CSWT did not reduce the degree of fibrosis in CSWT group compared with hypertension group. (**C**). In midmyocardium, induction of hypertension significantly increased the degree of fibrosis in hypertension and CSWT groups compared with control group. CSWT did not reduce the degree of fibrosis in CSWT group compared with hypertension group. (**D**). No significant difference of the degree of fibrosis was observed in epicardium among the three groups. Independent Student *t*-test was used.

**Figure 5 ijms-23-13274-f005:**
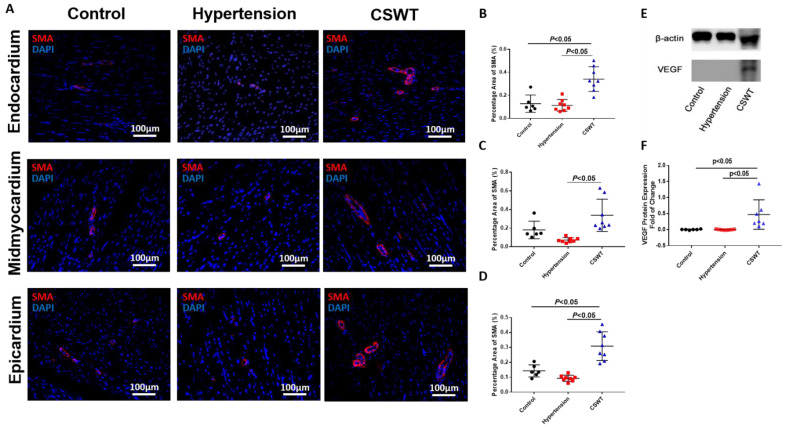
Angiogenesis effect of CSWT at the 20th week. Immunohistological staining of heart sections in the three groups at the 20th week. (**A**). Microvascular density in endocardium, midmyocardium and epicardium were assessed by immunofluorescence staining with SMA (smooth muscle actin) in the three groups. (**B**–**D**). There were no significant differences of positive−SMA staining in hypertension group compared with control group among three layers of myocardium. Additionally, CSWT significantly improved SMA−positive microvascular density compared with hypertension group in three layers of myocardium. (**E**). Representative Western blot analysis of VEGF (vascular endothelial growth factor) expression in three groups at the 20th week. (**F**). VEGF was significantly higher in CSWT group compared with control and hypertension groups. There was no significant difference of VEGF expression between control and hypertension groups. Independent Student *t*−test was used.

**Figure 6 ijms-23-13274-f006:**
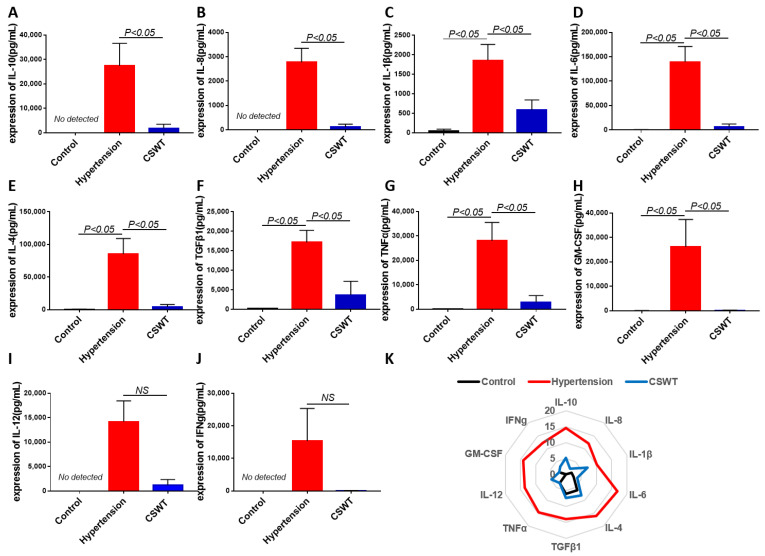
Anti-inflammatory effect of CSWT at the 20th week. The level of circulating cytokines was detected using cytokine array in the control, hypertension and CSWT groups (N = 4 for each group) at the 20th week. (**A**–**J**). After induction of hypertension, expression of IL-1β, IL-6, IL-4, TGFβ1, TNFα and GM-CSF significantly increased in hypertension group compared with control group, and the level of expression of IL-10, IL-8, IL-12 and IFNg was not detected in the control group. Expression of IL-10, IL-8, IL-1β, IL-6, IL-4, TGFβ1, TNFα and GM-CSF significantly decreased in the CSWT group compared with the hypertension group. There was no significant difference of expression of IL-12 and IFNg between hypertension group and CSWT group. (**K**). Expression of cytokines was detected using cytokine array in the control, hypertension and CSWT groups at the 20th week. Independent Student *t*-test and one-way ANOVA were used. NS: not significant.

## Data Availability

Data is available via reasonable request to corresponding authors.

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
