# Peer review of "Attenuation of Myocardial Dysfunction in Hypertensive Cardiomyopathy Using Non-R-Wave-Synchronized Cardiac Shock Wave Therapy"

_ijms, 2022, doi:10.3390/ijms232113274_

Round 1
Reviewer 1 Report
This work represents the first pre-clinical study which investigate the efficacy of CSWT -cardiac shock wave therapy, a non-invasive treatment in an experimental model. References must be up-dated
Author Response
Response to Reviewer 1 comments
This work represents the first pre-clinical study which investigate the efficacy of CSWT -cardiac shock wave therapy, a non-invasive treatment in an experimental model.
We thank the Reviewer for the comments that help us to improve this manuscript.
Point 1: References must be up-dated
Response 1: Thanks for the comments from the Reviewer. As recommended, we have updated revised the references and updated with some latest studies (Reference 1, 3, 5-7).
Reviewer 2 Report
This pre-clinical study evaluated the role of CSWT on hypertensive cardiomyopathy. The results indicate that CSWT has the ability to decrease LV diastolic disfunction and preserve LV systolic function in hypertensive cardiomyopathy. It is a well written manuscript with clearly presented methodology and results. I have no specific comments.
I have no specific comments with regards to the methodology and results presentation. In the introduction section the authors could add the incidence and prevalence values of heart failure in China, Europe and North America.
Perhaps the authors could further expand the discussion and provide an explanation with regards to the possible clinical implications of these findings. Which patients could benefit from CSWT? In which cardiac diseases the treatment would be shortened? This should be further specified in the Perspectives chapter. In addition, the authors could describe the protocol of a clinical study which would confirm these pre-clinical data? What would be inclusion and exclusion criteria? What would be the treatment duration?
Author Response
Response to Reviewer 2 comments
This pre-clinical study evaluated the role of CSWT on hypertensive cardiomyopathy. The results indicate that CSWT has the ability to decrease LV diastolic disfunction and preserve LV systolic function in hypertensive cardiomyopathy. It is a well written manuscript with clearly presented methodology and results. I have no specific comments.
We thank the Reviewer for the comments that help us to improve this manuscript.
Point 1: I have no specific comments with regards to the methodology and results presentation. In the introduction section the authors could add the incidence and prevalence values of heart failure in China, Europe and North America.
Response 1: As recommended, the incidence and prevalence of heart failure in China, Europe and North America has provide in the “Introduction Section” as follows:
“In China, up to 1.3% of population suffers from HF. The prevalence of HF in the United States and European vary from 1% to 12% based on different studies or reports.”
Point 2: Perhaps the authors could further expand the discussion and provide an explanation with regards to the possible clinical implications of these findings. Which patients could benefit from CSWT? In which cardiac diseases the treatment would be shortened? This should be further specified in the Perspectives chapter.
Response 2: Thanks for the comments from the Reviewer. We have revised and added the possible clinical implications of these findings in “Perspective Section”:
“Previous preclinical studies demonstrated that CSWT was a new therapy method for patients who suffered from chronic myocardial ischemia, refractory angina with no options of conventional treatment such as percutaneous coronary intervention (PCI) and coronary artery bypass grafting (CABG) [6]. According to above results, we supported previous studies which showed application of CSWT to induce angiogenesis and anti-inflammation to ameliorate CMD, which is a potential therapeutic target in HFpEF. Therefore, patients suffer from hypertensive cardiomyopathy with HFpEF could benefit from CSWT.”
Point 3: In addition, the authors could describe the protocol of a clinical study which would confirm these pre-clinical data? What would be inclusion and exclusion criteria? What would be the treatment duration?
Response 3: Thanks for the comments from the Reviewer. Our study demonstrated that CSWT can attenuate LV systolic and diastolic dysfunction via enhancement of myocardial neovascularization and anti-inflammatory effect in a large animal model of hypertensive cardiomyopathy with HFpEF. The safety and efficacy of CSWT in should be further investigated in clinical trials. Patients with hypertensive cardiomyopathy and evidence of with HFpEF will be eligible and recruit for this clinical trial. Exclusion criteria include patients with hypertrophic cardiomyopathy, HF post-myocardial infarction and hypertension with significant EF reduction.
The patients will undergo 2 or 3 sessions of CSWT over 3 months (once a month). Then the heart function will be measured during long-term follow-up.
Reviewer 3 Report
Does “CSWT group” mean hypertension + CSWT? This is not clear from the paper.
Authors also need to include “CSWT only group” to show the effects of CSWT on normotensive animals.
What is the N number for the Cytokine experiments?
The authors also need to show control for the Cytokine experiments.
Figure 6: The labels “P<0.05” and “P>0.05” are confusing.
Author Response
Response to Reviewer 3 comments
We thank the Reviewer for the comments that help us to improve this manuscript.
Point 1: Does “CSWT group” mean hypertension + CSWT? This is not clear from the paper.
Response 1: Thanks for the comments from the Reviewer. We are sorry about this misunderstanding. Yes, “CSWT group” means hypertensive + CSWT. We have revised the description in “Methods Section”:
“At 10 weeks following Ang-II infusion and DOCA implantation, animals with hypertensive cardiomyopathy were assigned to no therapy group (hypertension group, n=8) or CSWT group (n=8).”
Point 2: Authors also need to include “CSWT only group” to show the effects of CSWT on normotensive animals.
Response 2: Thanks for the suggestion from Reviewer. However, due to the breakout of COVID-19 in Hong Kong, the large animal services and operation room has been closed and we could not perform any pig study in a short time. We would perform the next step of study related with CSWT and include” CSWT only group” in the future.
Point 3: What is the N number for the Cytokine experiments?
Response 3: Thanks for the comments from the Reviewer. N number for the cytokine array is 4 for each group. We had revised and described the sample size in the manuscript.
Point 4: The authors also need to show control for the Cytokine experiments.
Response 4: Thanks for the suggestion from Reviewer. As recommended, the cytokine experiment of control group has been provided in Figure 6, Graphical Abstract and the “Results Section” (Cytokines array data section):
“The level of circulating cytokines was detected using cytokine array in the control, hypertension and CSWT groups at the 20th week (Figure 6). After induction of hypertension, expression of IL-1β, IL-6, IL-4, TGFβ1, TNFα and GM-CSF significantly increased in hypertension group compared with control group and the level of expression of IL-10, IL-8, IL-12 and IFNg was not detected in the control group (Figure 6A-J). Expression of IL-10, IL-8, IL-1β, IL-6, IL-4, TGFβ1, TNFα and GM-CSF significantly decreased in the CSWT group compared with the hypertension group (Figure 6A-H). There was no significant difference of expression of IL-12 and IFNg between hypertension group and CSWT group. (Figure 6 I & J).”
Point 5: Figure 6: The labels “P<0.05” and “P>0.05” are confusing.
Response 5: Thanks for the comments from the Reviewer. We have revised the Figure 6, using “NS” to instead of “P>0.05” to represent no difference.
Round 2
Reviewer 3 Report
The inclusion of “CSWT only group” is particularly important for Fig. 5. Authors indicates that due to the breakout of COVID-19 in Hong Kong, the large animal services and operation room has been closed. Please at least discuss that possibility of CSWT doing things in normotensive animals.
Author Response
Response to Reviewer 3 comments
We thank the Reviewer for the comments that help us to improve this manuscript.
Point 1: Please at least discuss that possibility of CSWT doing things in normotensive animals.
Response 1: Thanks for the comments from the Reviewer. Our study did not include a control group with the CSWT treatment. Because in normotensive animals, there is no evidence that shock wave therapy induces cellular damage and inflammatory response in the myocardium and change the ultrastructure morphology, blood pressure and cardiac ventricular function when used within a therapeutic range. We have revised the “Discussion Section”:
“Prior studies showed that CSWT improve cardiac angiogenesis in ischemic cardiac disease without any increase of serum Troponin I level and cardiac arrhythmic risk[23]. Meanwhile, in normotensive animals, there is no evidence that shock wave therapy induces cellular damage and inflammatory response in the myocardium and change the ultrastructure morphology, blood pressure and cardiac ventricular function when used within a therapeutic range [25]. Our results support these findings and further demonstrated that CSWT did not cause any complication in hypertensive cardiomyopathy. Our study showed that no increase of blood pressure, no in-crease of serum Troponin T as well as no cardiac arrhythmia and cardiac sudden death were observed.”
